# The Post-Effects of the Authenticity of Rural Intangible Cultural Heritage and Tourists’ Engagement

**DOI:** 10.3390/bs14040302

**Published:** 2024-04-05

**Authors:** Yifan Zuo, Tianning Lan, Shuangshuang Liu, Hongfa Zeng

**Affiliations:** 1School of Sports, Shenzhen University, Shenzhen 518061, China; yifanzuo@szu.edu.cn (Y.Z.); 2300371016@email.szu.edu.cn (S.L.); 2Shenzhen Tourism College, Jinan University, Shenzhen 518053, China; lantianning2555@163.com

**Keywords:** rural intangible heritage tourism, authenticity, tourist engagement, destination image, loyalty

## Abstract

Amidst the burgeoning interest in rural Intangible Cultural Heritage (ICH) tourism, this study, anchored in the Stimulus–Organism–Response (S–O–R) theoretical framework, chose Meizhou Island in Fujian Province as a case study to examine the impacts of tourists’ perceptions of authenticity and their engagement levels on the destination image and loyalty towards rural ICH tourism. Utilizing Structural Equation Modeling (SEM) to analyze survey data, findings revealed that higher perceptions of a destination’s preservation of original characteristics and traditional values correlate with a more favorable overall perception of the destination, albeit with smaller impact on emotional connections. Conversely, depth of tourist engagement was shown to enhance both understanding and emotional bonds with the destination. Further, familiarity with the destination was found to foster affection, thereby increasing the likelihood of repeat visits or recommendations. The more tourists know about a place, perceiving it as authentic, the more likely they are to remain loyal; similarly, deeper engagement enhances understanding and affection, increasing the probability of revisiting or recommending the place. These outcomes not only offer new insights into tourist behavior but also provide theoretical and practical guidance for the protection and development of rural ICH tourism, destination marketing, and management strategies, thus promoting the sustainable development of rural ICH tourism.

## 1. Introduction

The core concept of intangible cultural heritage (hereafter referred to as “ICH”) protection and inheritance is the preservation and continuation of “living history”. Through this preservation and continuation, it is transitioned from memory to experience [1]. On one hand, the process of integrating ICH into modern life promotes cultural heritage. On the other, it drives industrial development, especially during the “14th Five-Year Plan” period, where ICH plays a significant role in rural revitalization and poverty alleviation. Villages that rely on ICH resources actively encourage the participation of the impoverished population. This participation enables them to benefit from various ICH-related industries, such as handicrafts, performances, and rural tourism [1]. However, rural ICH tourism currently faces a severe issue: tourists show great interest in attractions and activities but lack understanding and awareness of the forms and destinations of rural ICH tourism [2]. This vague image perception often affects tourists’ travel decisions and thus weakens the competitiveness of the rural ICH tourism market and limits its development.

ICH in tourism embodies four characteristics: authenticity, experientiality, sustainability, and inimitability. Authenticity refers to the extent to which ICH maintains its original features and traditional values. The substitutive relationship issue between the protection of authenticity and participatory development in tourism raises questions about whether there exists a conflict between authenticity protection and commercial development, and whether these two can adapt and adjust to each other effectively to enhance tourist loyalty for mutual development.

Previous research indicates that tourism development is a double-edged sword for ICH protection. Blind tourism development can distort the authenticity of ICH, but the effective cultural reproduction of authenticity benefits the quality of tourists’ experience and the perceived value [3]. In heritage tourism experiences, tourists’ primary motivation and core experience need is to feel the authenticity of ICH culture. This perception often further influences tourists’ loyalty behaviors, such as revisits and recommendations [4]. The relationship between authenticity perception and tourist loyalty has been extensively validated [5]. In recent years, channels and groups for the revitalization of ICH have become more diverse. Tourists gain from their participation in the tourism experience of ICH content in terms of perception, learning, and enjoyment, and this participation has gradually become an effective means of fostering ICH revitalization in tourist destinations.

Generally speaking, tourist participation refers to the emotional, cognitive, and behavioral engagement of tourists in the travel experience. Understanding tourist participation can effectively improve the predictability of tourist behavior [6]. Tourist engagement has been proven to be a precursor to tourist loyalty [7]. As a significant psychological representation of tourists’ perception of the destination, destination image refers to the overall opinions and impressions that an individual or group holds about a tourism destination. In rural ICH tourism, the destination image plays a crucial role in influencing tourists’ travel decisions and post-travel evaluations, serving as a key antecedent to loyalty [8]. Tourist loyalty primarily indicates tourists’ tendency to revisit a specific tourism destination, service, or brand and their willingness to continue choosing and recommending it to others in the future [5]. Tourism destinations striving to integrate ICH into new attractions often end up in disharmony with the surrounding environment and fall into the development cycle described by Butler, which eventually evolves into highly fictionalized and commercialized tourist attractions.

The synthesis of the above research findings indicates that authenticity protection and tourist engagement have always been important concepts in the market development and marketing of rural ICH tourism. Research on rural ICH tourist loyalty has shifted from causation to the exploration of the mechanisms of influence. However, previous studies have been confined to a fixed model framework of authenticity, satisfaction, and loyalty, with limited focus on tourist engagement and destination image, despite their significant impact on rural ICH tourist loyalty. Currently, there is a lack of research on authenticity protection and participatory development in rural ICH tourism. The cognitive and emotional aspects of destination image in the formation mechanism of tourist loyalty are overlooked. In light of this, the present study, which is based on the stimulus–organism–response (S–O–R) theoretical model, employs a questionnaire survey method. The study uses Meizhou Island in Fujian as a case study and aims to construct a structural model of perceived authenticity, tourist engagement, destination image, and tourist loyalty. The intention is to enhance understanding of the drivers of loyalty among rural ICH tourists and to provide new insights and references for the reexamination and innovation of the market development and marketing strategies of rural ICH tourism. As an innovative aspect of the integration of culture and tourism, rural ICH tourism should emphasize its cultural essence and consider whether a balance can be achieved between revitalization protection and commercial development, in order to realize the high-quality, collaborative development of culture and tourism [1].

## 2. Literature Review and Hypothesis Development

### 2.1. Conceptual Definition

#### 2.1.1. Authenticity Perception

Authenticity is a crucial concept in the field of world heritage conservation. Initially used by scholars in the study of museums and cultural heritage, it denotes the genuineness of exhibits or cultural relics [9]. In the 1970s, MacCannell [8] formally introduced the term “authenticity” into the field of tourism sociology to study tourists’ motivations and experiences at historical and cultural sites. In tourism studies, the concept of authenticity has evolved into various perspectives, such as objective authenticity, constructivist authenticity, and customized authenticity. Customized authenticity, in particular, considers authenticity to be co-constructed as it is based on the interaction between host and guest (the tourism destination and the tourists). The core idea is that tourists’ authenticity needs towards an object are formed in advance by internal or external factors, and the host can provide tourism activities and products that meet their imagination and needs, thereby co-creating authentic tourism experiences with tourists [10]. Therefore, by considering tourism and heritage conservation research to be fundamentally based on the interaction between tourists and hosts, this paper adopts the perspective of customized authenticity to define the authenticity perception of rural intangible cultural heritage tourists. That is, tourists can form authenticity needs towards an object that are influenced by internal or external factors, and the host can provide tourism activities and products that meet their imagination and needs. Together, they can create authentic tourism experiences [10].

Currently, many countries, including China, have identified the experience of authenticity as a key element in the future development of heritage tourism [11]. Some scholars believe that compared to the authenticity of the object, human perception of authenticity plays an even more significant role in promoting the development of heritage tourism [11]. As a key antecedent of tourist loyalty, the exploration of the impact of authenticity perception on tourist loyalty behavior is a common research paradigm. However, the existing studies tend to focus more on the relationship between satisfaction or place attachment and tourist loyalty, overlooking the fact that other psychological representational elements can also play a crucial role.

#### 2.1.2. Tourist Engagement

Tourist engagement, which developed from the important concept of “Customer engagement” in the fields of marketing and service management, differs from the concepts of “Participation” and “Involvement”. Tourist engagement, as translated from “Visitor engagement”, places more emphasis on the heightened experiences and interactions during the tourism process. It can manifest as an interest in or commitment to participation and can also be seen as a component of behavior or as a result of the activities engaged in [12]. Compared to involvement, engagement establishes a more positive consumer relationship with specific engagement objects and involves a higher level of behavioral commitment [13]. Therefore, this study defines tourist engagement as the engagement of tourists in and their commitment to their travel experiences.

The concept of engagement has not received much attention in the field of tourism, and most of the few studies on tourist engagement have been conceptual [14]. It was not until Taheri et al. [15] developed a participation scale in the context of museums that research on the antecedents and consequences of tourist engagement began to increase. For example, studies have shown that the higher the degree of tourist engagement, the more memorable the tourism experience, leading to unforgettable tourism memories [16]. Leisure motivation and knowledge level are antecedent variables of tourist engagement, while word of mouth and the intention to revisit are outcome variables [17]; tourists’ satisfaction and memories are antecedents affecting tourist engagement [18].

#### 2.1.3. Destination Image 

Destination Image is a core concept in the field of tourism destination marketing. Hunt [19], who first defined the concept of destination image, emphasized that it focuses on the stream of subjective perception and defined it as the sum of beliefs, perceptions, and impressions held by an individual or a group about a place or destination.

The image of a destination is a mature construct with a complex structure, and scholars have diverse opinions on its composition, with no unified understanding yet formed. Some scholars have proposed a “cognitive-affective-conative” three-dimensional structure of destination image from the perspective of tourists’ psychological perceptions, where the conative image emphasizes the destination image that drives tourists to engage in specific tourism behaviors. Thus, it is more akin to a behavioral inclination and is therefore not commonly used as a dimension in applied research [20]. Consequently, a two-dimensional structure of “cognitive-affective” destination image has been formally proposed, where the cognitive image emphasizes tourists’ perception of destination attribute information, and the affective image is the emotional evaluation of tourists towards the destination attributes and the surrounding environment.

#### 2.1.4. Tourist Loyalty

The concept of loyalty is the deep-seated commitment to repurchasing, revisiting, or preferentially choosing a product or service in the future, even when environmental conditions and marketing efforts change. This loyalty leads to long-term repetitive purchases of the same brand or series of products or services [21,22]. Tourist loyalty often manifests as repetitive behaviors and attitudes towards the same travel destination, word-of-mouth recommendations, and a tendency to revisit [22]. Compared to consumer behavior, the concept of tourist loyalty in the tourism industry is more complex. On one hand, in contrast to specific products and services, travel involves longer durations, greater distances, and significant expenses and is often influenced by financial status, holidays, traveler health, and destination capacity, which can hinder the development of tourist loyalty [23]. On the other hand, tourists are motivated by the pursuit of novel experiences [24]. The core of “behavioral loyalty” in tourist loyalty is reflected in the revisiting of the same destination, which contradicts the desire for novelty and repetitive travel intentions. Therefore, this study incorporates the measurement of attitudinal loyalty while measuring tourist loyalty and adopts the concept of experiential loyalty from McKercher et al. [25]. To some extent, experiential loyalty utilizes the substitutability and similarity of travel experiences and quality. It is hypothesized that tourists may remain loyal to the same style, experience, and environment of travel, i.e., different locations but the same type of destinations. In particular, loyalty to intangible experiences, such as rural intangible cultural heritage tourism, is more likely to shift.

### 2.2. Research Hypotheses

This study adopts the “S–O–R” framework as its theoretical basis. Specifically, in the context of a destination, the perceived attributes of the destination (such as scenery, culture, etc.), the behavioral outcomes (such as participation in activities and product and service experiences), and the experiential values (such as pleasure, culture, education, etc.) all constitute external stimuli. In the context of rural intangible cultural heritage tourism, the perception of authenticity is an environmental stimulus of the destination, and tourist engagement is an external stimulus related to the experience of tourism products or services. The perception of authenticity and tourist engagement, as variables of external stimuli, can trigger tourists’ psychological activities and emotional states, thereby motivating the corresponding behavioral responses.

#### 2.2.1. The Relationship between Perceived Authenticity, Tourist Engagement, and Destination Image

For tourists engaging in rural intangible cultural heritage tourism, the authenticity of the travel experience is often a key factor that needs to considered. The perception of cultural authenticity is the main motivation and experiential need of tourists in the heritage tourism market [4]. High-quality authentic experiences often contribute to the formation of a positive destination image. For example, studies have found that the perception of authenticity can promote a positive image of historical districts as destinations [26]. The perception of authenticity can positively affect tourists’ travel experiences, thereby forming a positive destination image. The impact of perceived authenticity on destination image has been validated in the context of ancient village tourism [27]. Hence, the following hypotheses are proposed:

**Hypothesis** **1.**
*Tourists’ perception of authenticity has a significant positive impact on destination image.*


**H1a.** 
*Tourists’ perception of authenticity has a significant positive impact on cognitive image.*


**H1b.** 
*Tourists’ perception of authenticity has a significant positive impact on affective image.*


Tourist engagement is the interaction between tourists and the destination. This interaction, which is a psychological or behavioral representation of the tourist experience, can enhance positive perceptions and emotions towards the destination and contribute to a favorable destination image [16]. Consequently, the level of interaction between tourists and the destination positively influences the perception of their travel experience and the destination image [28]. Studies using qualitative methods have demonstrated that tourist engagement can positively impact the image of heritage destinations [6]. The following hypotheses are proposed:

**Hypothesis** **2.**
*Tourist engagement significantly positively affects the destination image.*


**H2a.** 
*Tourist engagement significantly positively affects the cognitive image.*


**H2b.** 
*Tourist engagement significantly positively affects the emotional image.*


#### 2.2.2. The Relationship between Destination Image and Loyalty

According to the S-O-R theory, tourist loyalty can be seen as a natural response that is influenced by external stimuli and psychological factors. The destination image, as a precursor variable to tourist loyalty, has been extensively validated in various research contexts, theories, and cases and is often conceptualized as a unidimensional construct or cognitive image. For example, the positive impact of tourists’ perceived cognitive image of the destination on loyalty has been empirically tested from the perspective of tourist perception [29]. The research that views the destination image as a multidimensional construct occupies only a small part of the studies. The studies on multidimensional structural variables offer a deeper and more comprehensive understanding of the relationships between variables. Hernández-Lobato et al. [30], based on the premise that the destination image comprises cognitive–emotional dimensions, confirmed that the entertainment attribute of the cognitive image and the emotional image significantly positively affect tourist loyalty. Additionally, with regard to the relationship between cognitive and emotional images, there are two differing views: “emotion affects cognition” and “cognition affects emotion”. Given that “cognition affects emotion” was more widely recognized and applied in past tourism research [31], the following hypotheses are proposed:

**Hypothesis** **3.**
*The cognitive image significantly positively affects the emotional image.*


**Hypothesis** **4.**
*The destination image significantly positively affects loyalty.*


**H4a.** 
*The cognitive image significantly positively affects loyalty.*


**H4b.** 
*The emotional image significantly positively affects loyalty.*


#### 2.2.3. The Relationship between Authenticity Perception, Tourist Engagement, and Loyalty

In heritage tourism, studies on the impact of authenticity perception and tourist engagement on tourist loyalty are common research models. For instance, studies have confirmed the direct impact of authenticity perception on tourist loyalty, with authenticity perception influencing loyalty through existential authenticity [32]. Other studies have found that tourist engagement in heritage tourism sites positively affects tourists’ loyalty to the destination [7]. Although the impacts of authenticity perception and tourist engagement on tourist loyalty have been validated by numerous scholars, the integration of authenticity perception, tourist engagement, and tourist loyalty into a single context for discussion and verification remains valuable, especially in rural intangible heritage tourism. Rural intangible heritage tourism, as an innovation in cultural and tourism integration during the “14th Five-Year Plan” period, faces the pressing issue of the need to balance authenticity preservation with tourist engagement. Based on this, the following hypotheses are proposed: 

**Hypothesis** **5.**
*Tourists’ perception of authenticity has a significant positive impact on loyalty.*


**Hypothesis** **6.**
*Tourist engagement significantly positively affects loyalty.*


#### 2.2.4. The Mediating Role of Destination Image

Considering the previously discussed relationship between authenticity perception, tourist engagement, and destination image, as well as the S–O–R theoretical framework, the destination image, as a psychological representation element, is aptly positioned as the “Organism (O)” mediating variable. It is reasonably hypothesized that the destination image mediates the impact of authenticity perception and tourist engagement on tourist loyalty. Research has demonstrated that the destination image can serve as a mediator between authenticity and other variables [33]. Although no study has empirically tested the destination image as a mediator for tourist engagement, Su et al. [6] directly verified the relationship between tourist engagement, heritage site image, and satisfaction, with satisfaction being a key factor affecting loyalty. Additionally, in studies where the destination image acts as a mediator, few have delved into whether cognitive and emotional images can independently serve as variables with mediating effects. Hence, the following hypotheses are proposed:

**Hypothesis** **7.**
*The destination image mediates between authenticity perception and loyalty.*


**H7a.** 
*The cognitive image mediates between authenticity perception and loyalty.*


**H7b.** 
*The emotional image mediates between authenticity perception and loyalty.*


**H7c.** 
*The cognitive–emotional image mediates between authenticity perception and loyalty.*


**Hypothesis** **8.**
*The destination image mediates between tourist engagement and loyalty.*


**H8a.** 
*The cognitive image mediates between tourist engagement and loyalty.*


**H8b.** 
*The emotional image mediates between tourist engagement and loyalty.*


**H8c.** 
*The cognitive–emotional image mediates between tourist engagement and loyalty.*


Based on the above analysis and hypotheses, the individual tourist characteristics are considered as control variables to construct a conceptual research model (Figure 1).

## 3. Research Design

### 3.1. Overview of the Case Study Area

In recent years, Meizhou Island has seen a significant boost in its tourism economy, driven by the Mazu culture and faith, as well as the island’s natural environment. The total number of tourists and the tourism revenue have been continuously rising. In the initiative of “Creating a Civilized City and Building Beautiful Villages”, Meizhou Island emphasizes supporting the cultural tourism industry, urging the public to participate in and promote the activities of the beautiful village and rural revitalization. The government has integrated the island’s intangible cultural heritage resources with rural tourism, focusing on creating “one village, one product, one landscape, one street”. This effort aims to improve the sanitary, tourism, and ecological environments of each village and highlights the industrial development advantages of achieving rural revitalization.

Meizhou Island is located in Xiuyu District, Putian City, Fujian Province, China. It covers a land area of 14.35 km^2^ and has a population of 40,500. It comprises 11 administrative villages under Meizhou Town, including Gongxia Village, Dongcai Village, and others. The island is famous for its Mazu belief. In 2009, the Mazu belief was inscribed on the Representative List of the Intangible Cultural Heritage of Humanity, and the following year the island was listed as a National 5A Scenic Area. Currently, Meizhou Island boasts over 30 scenic spots, such as the Mazu Ancestral Temple and the Mazu Culture Film and Television Park.

### 3.2. Measurement Tools and Questionnaire Design

The main part of the survey questionnaire consists of four sections: the measurement scales for the antecedent variables of authenticity perception and tourist engagement, destination image, loyalty, and a demographic factors survey.

The measurement of authenticity perception is based on the scale used by Lu et al. [26] in their studies. It consists of eight indicators. The items were translated from English to Chinese, and teachers and students who had visited Meizhou Island were invited to test the questionnaire. The test was followed by adjustments and exclusions based on their feedback. The measurement of tourist engagement refers to the scale developed by Taheri [15], which includes eight indicators. Some of the expressions of tourist engagement behaviors are modified to reflect intentions. These adjustments align with the concept of engagement, which involves tourists’ engagement, commitment, and interest in tourism experiences [34]. The destination image is generally composed of cognitive and affective images. Thus, the measurement of the cognitive image draws on the scales designed by Chi et al. [35] and Chen et al. [36]. These scales comprise five dimensions and 17 indicators. The affective image is measured using a five-point differential scale developed by Russell [37], which includes four sets of adjectives. The loyalty measurement primarily focuses on behavioral intention items, drawing on scales by Chen et al. [38] and Kolar et al. [39], with five indicators, using a Likert five-point scale.

After the preliminary formation of the questionnaire, experts were invited to assess and revise the descriptions to finalize the pre-survey questionnaire. The pre-survey was conducted from 26–29 October 2023, on Meizhou Island, using convenience sampling for on-site distribution. A total of 127 valid questionnaires were collected. To test the internal consistency of the questionnaire, the Cronbach’s α coefficient method was applied to the overall and individual scales, with “Corrected Item-Total Correlation (CITC)” and “Cronbach’s α if Item Deleted” used for purification. As a result, one item from each of the scales—the authenticity perception scale, tourist engagement scale, and affective image scale—was removed, for a total of three deleted items. Thus, the final questionnaire was formed.

### 3.3. Formal Questionnaire Distribution and Sample Overview

The formal survey was conducted from 11–15 December 2023, on Meizhou Island, Fujian Province. Non-probability convenience sampling was used for simultaneous online and offline distribution, with the online questionnaires distributed through Questionnaire Star along with the offline paper questionnaires. A total of 436 questionnaires were distributed on-site, with 31 invalidated due to incomplete responses, resulting in 405 valid offline questionnaires. Online, 71 questionnaires were distributed, with nine invalidated as the participants had never visited Meizhou Island, yielding 62 valid online responses. In total, 507 questionnaires were distributed, with 40 invalidated, leading to 467 valid responses. The response rate was 92.11%.

From 11–15 December 2023, a formal survey was conducted on Meizhou Island, Fujian Province. Given the research objectives and resource constraints, this study employed a non-probability convenience sampling method, distributing questionnaires both online and offline. This sampling method ensured the effective collection of data from the target population, with specific operations as follows: (1) The research team distributed paper questionnaires at various locations on Meizhou Island, including tourist attractions, hotels, and other public places, to cover as many tourists as possible. A total of 436 questionnaires were distributed on-site, with 31 deemed invalid due to incomplete responses, resulting in 405 valid questionnaires collected offline. (2) Concurrently, the research team created an online questionnaire using Questionnaire Star (https://www.wjx.cn, accessed on 11 December 2023) and invited tourists who had completed the paper questionnaire to recommend the online survey to others. A total of 71 questionnaires were filled out online, with nine invalidated for never having visited Meizhou Island, leaving 62 valid questionnaires collected online. In total, 507 questionnaires were distributed both online and offline, with 40 invalidated, resulting in 467 valid questionnaires collected, making for a 92.11% effective recovery rate.

## 4. Research Results

### 4.1. Sample Characteristics

The survey sample comprised slightly more female than male tourists. The tourists were predominantly young and middle-aged, with no significant income disparities and a moderate overall education level. Bachelor’s degree holders constituted 27.8% of the sample. Nearly 40% had some understanding of Mazu culture (Table 1).

### 4.2. Scale Testing

The SPSS AMOS 21.0 software was used for confirmatory factor analysis to test the structural relationships of the scales (Table 2). The model fit indicators met the general research standards: χ^2^ = 850.416, df = 687, χ^2^/df = 1.238 (<3). RMSEA = 0.041 (<0.08), GFI = 0.915 (>0.9), IFI = 0.981 (>0.9), TLI = 0.979 (>0.9). The standardized factor loadings (λ), composite reliability (CR), and average variance extracted (AVE) for each scale exceeded the standard values, indicating good convergent validity. The overall (0.930) and individual scale Cronbach’s α were above the threshold, demonstrating good internal consistency. The square root of each factor’s AVE was greater than the standardized correlation coefficients outside the diagonal, confirming the study’s discriminant validity.

### 4.3. Structural Model Testing

Using AMOS 21.0 software and the maximum likelihood method, the initial conceptual model with five first-order latent variables was estimated. The results are summarized in Table 3. The analysis showed that the standardized path coefficient of authenticity perception to affective image was 0.068, with a C.R. value of 1.179, which was less than 1.96, and a *p*-value of 0.238, which was greater than 0.05, indicating no significant impact of authenticity perception on affective image. Thus, hypothesis H1b was not supported. Aside from this insignificant path, all the other paths showed significant impacts. Therefore, to achieve a better model effect, this path was deleted in the model revision.

Following the deletion of the “authenticity perception → affective image” path, the reanalysis resulted in the model shown in Figure 2. The model fit indicators were: χ^2^ = 450.313, df = 315, χ^2^/df = 1.430 (<3). RMSEA = 0.030 (<0.08), GFI = 0.933 (>0.9), AGFI = 0.919 (>0.9), NFI = 0.926 (>0.9), IFI = 0.977 (>0.9), TLI = 0.974 (>0.9), CFI = 0.976 (>0.9), indicating a good overall model fit with the sample data.

The structural equation model path coefficients are presented in Table 4. Regarding the path analysis of authenticity perception with destination image dimensions, the authenticity perception had a significant positive impact on cognitive image, with a standardized path coefficient of 0.319 and a *p*-value of less than 0.001, supporting hypothesis H1a. Regarding the impact of tourist engagement on the destination image dimensions, tourist engagement significantly positively influenced both the cognitive (β = 0.257, *p* < 0.001) and the affective (β = 0.271, *p* < 0.001) images, supporting hypotheses H2a and H2b. In the two-dimensional structure of destination image, the cognitive image positively influenced the affective image (β = 0.312, *p* < 0.001), supporting hypothesis H3. Regarding the path analysis results for the relationships between the antecedent variables and the outcome variable, both authenticity perception (β = 0.140, *p* < 0.001) and tourist engagement (β = 0.156, *p* < 0.001) exhibited significant positive impacts on tourist loyalty, indicating the support for hypotheses H5 and H6. In terms of the path analysis results for the destination image dimensions’ impact on tourist loyalty, cognitive image (β = 0.284, *p* < 0.001) and affective image (β = 0.313, *p* < 0.001) both significantly positively influenced tourist loyalty, supporting hypotheses H4a and H4b.

### 4.4. Mediation Test

This study employed AMOS 21.0 to implement a bootstrap program to verify the mediating effect of the model. As the basic premise for the validation of the mediation of variables is the significant impact of the independent variable on the dependent variable, this study does not validate the mediating role of the emotional image between tourists’ perceived authenticity and loyalty. Using the bootstrap method in AMOS 21.0 and running it 5000 times, the bias-corrected and percentile levels at a 95% confidence interval were obtained. The results of the mediation effect test are shown in Table 5.

The study found that the total effect of perceived authenticity on loyalty is (β = 0.316, bias-corrected 95% CI: [0.152, 0.373]). The indirect effect of perceived authenticity on loyalty mainly passes through two paths: “Perceived Authenticity → Cognitive Image → Loyalty” (β = 0.091, bias-corrected 95% CI: [0.045, 0.163]) and “Perceived Authenticity → Cognitive Image → Emotional Image → Loyalty” (β = 0.031, bias-corrected 95% CI: [0.013, 0.069]). The direct effect of perceived authenticity on loyalty is (β = 0.140, bias-corrected 95% CI: [0.037, 0.239]).

The total effect of tourist engagement on loyalty is (β = 0.339, bias-corrected 95% CI: [0.234, 0.454]). The indirect effect of tourist engagement on loyalty mainly passes through three paths: “Tourist engagement → Cognitive Image → Loyalty” (β = 0.073, bias-corrected 95% CI: [0.035, 0.145]), “Tourist engagement → Emotional Image → Loyalty” (β = 0.085, bias-corrected 95% CI: [0.036, 0.156]), and “Tourist engagement → Cognitive Image → Emotional Image → Loyalty” (β = 0.025, bias-corrected 95% CI: [0.010, 0.055]). The direct effect of tourist engagement on loyalty is (β = 0.156, bias-corrected 95% CI: [0.065, 0.252]).

## 5. Conclusion and Discussion

### 5.1. Conclusion

In the context of rural intangible cultural heritage tourism, perceived authenticity can significantly positively affect tourists’ cognitive destination image. Simultaneously, tourist engagement can significantly positively influence tourists’ cognitive and emotional destination images. When tourists perceive the originality and authenticity of rural intangible cultural heritage during their travel experience, they are more likely to form positive cognitive images and favorable impressions of various aspects of the destination, such as its natural scenery, transportation environment, cultural resources, safety, entertainment, and value for money. Furthermore, when tourists actively interact with local residents or objects or are committed to participating/willing to participate in rural intangible cultural heritage tourism activities, they develop positive emotional and cognitive perceptions of the destination.

The “cognitive-emotional” dual structure of the destination image is validated again. On one hand, the cognitive image can significantly positively affect the emotional image. On the other hand, the five dimensions of cognitive image proposed in this study—experiential environment, leisure and shopping, cultural environment, auxiliary facilities, and value and service—can serve as a benchmark for the assessment of the cognitive image of rural intangible cultural heritage tourism destinations.

In rural intangible cultural heritage tourism, tourists’ perceived authenticity positively affects their loyalty, directly or indirectly, through the mediating effect of the destination’s cognitive image. The “cognitive-emotional” image serves as a chained mediator between perceived authenticity and tourist loyalty. Furthermore, tourist engagement significantly positively affects tourist loyalty, with destination image partially mediating between tourist engagement and loyalty. Tourist engagement can also indirectly positively influence tourist loyalty through the “cognitive-emotional” image.

### 5.2. Discussion

#### 5.2.1. The Mediating Role of Destination Image

As the results indicate, perceived authenticity can positively impact tourists’ cognitive destination image. That is, when tourists perceive the originality and authenticity of a heritage site during their travel experience, they are more likely to form positive cognitive images and favorable impressions about various attributes of the destination and its information and environment. These results are consistent with the findings of Lu et al. on Chinese historical districts [26]. However, tourists’ perception of authenticity does not have a significant impact on the emotional image of the destination. Although some studies have found a relationship between the perception of authenticity and the destination image, this causal relationship is only focused on the perception of authenticity and the unidimensional destination image. Research is lacking on the relationship between the perception of authenticity and the multidimensional destination image. Tourist engagement can positively influence both the cognitive and emotional image of the destination, with a greater impact on the emotional image than on the cognitive image. In other words, if tourists actively interact with people or objects at the tourist site or are committed to participation or willing to participate in activities at the tourist destination, they will form positive emotions and cognitive images with regard to the destination. This finding differs from the results of Su et al., where the heritage destination image was not affected by tourist engagement [6]. This suggests that the impact of tourist engagement on the destination image may vary depending on the specific heritage tourism destination. From the perspective of path coefficients, the perception of authenticity has a greater positive effect on tourists’ cognitive image of the destination than tourist engagement. Therefore, enhancing tourists’ perception of authenticity is an effective way to improve their cognitive image of the destination, and strengthening tourists’ interaction with and engagement in the destination is an effective way to enhance their emotional and overall image of the destination. This study provides empirical evidence for the interrelationship between authenticity perception, tourist engagement, and the two-dimensional structure of the “cognitive-affective” destination image in the context of rural intangible cultural heritage tourism. Moreover, with regard to the internal composition and influence of the destination image, the cognitive image can significantly positively influence the emotional image. This result supports the assertions of some scholars and revalidates the two-dimensional structure of the destination image in the context of rural intangible cultural heritage tourism. The dimensions of the cognitive image proposed in the study—experiential environment, leisure and shopping, cultural environment, auxiliary facilities, value, and service—can serve as a benchmark for the assessment of the cognitive image of rural intangible cultural heritage tourism destinations [40].

Tourists’ perception of authenticity not only has a significant direct impact on loyalty but also indirectly influences tourists’ loyalty through the impact on the destination’s cognitive image and “cognitive-affective” image. In other words, when tourists perceive the authenticity of the heritage tourism destination, they can form a positive cognitive image of the destination, followed by a positive emotional image, which ultimately enhances their willingness to revisit, their word-of-mouth recommendations, and their desire for similar types of tourism experiences. Currently, disregarding the dimension of authenticity perception, a general consensus has been reached in tourism studies regarding the direct relationship between authenticity perception and loyalty [32]. Regarding their indirect relationship, many scholars have found that concepts such as satisfaction, place attachment, and involvement can play a mediating role between authenticity perception and loyalty. However, no studies have found that authenticity perception can indirectly affect tourist loyalty through the destination image. This study focuses on a more practical factor, the destination image, and verifies that the cognitive image and “cognitive-affective” image can serve as partial mediators between the two. The “cognitive-affective” image is a chain mediator between authenticity perception and tourist loyalty. According to the standardized effect values, the direct impact of authenticity perception on tourist loyalty is significantly greater than the indirect impact of any one path. However, the combined indirect impacts of the two paths are close to the direct impact of authenticity perception on tourist loyalty.

#### 5.2.2. Management Implications

Regarding the cultivation of tourist loyalty, for managers of rural intangible cultural heritage tourism the achievement of a proper balance between the protection of authenticity and tourist engagement can lead to the sustainable development of rural intangible cultural heritage tourism and the realization of rural revitalization. In the integration of the conclusions of this paper, from the perspective of protecting the authenticity of intangible cultural heritage and tourism development, this study proposes that there are management service implications for the promotion of the loyalty of tourists to rural intangible cultural heritage tourism:The study found that some rural intangible cultural heritage (ICH) areas experience the turbulence of economic and protective game playing in the process of inheritance and development, becoming commercialized, crude, and formalized under the temptation of quick benefits, thereby diminishing their precious heritage value and deviating from the authentic folk faith culture [41]. This negative development phenomenon leads to tourists perceiving a lack of authenticity, which severely affects the destination image construction and tourist loyalty. For rural ICH tourism, it is necessary to emphasize the cultural characteristics of ICH authenticity to attract tourists, as authenticity is the core attribute and principle for sustainable rural ICH tourism. As the birthplace or heritage site of ICH, it is essential to carefully protect and pass on related resources (such as cultural beliefs, customs, folk activities, and material heritage), delve into the cultural connotations of ICH, focus on the authenticity of the rural environment and ICH resources, and provide tourists with authentic content experiences, fostering interaction between tourists and the destination. Tourism functional areas can be divided according to their positioning and can implement diversified tourism development strategies to prevent other types of tourism from infiltrating rural ICH tourism and ruining the authentic experience for tourists;Additional interactive experience projects should be established to promote tourist engagement. Rural ICH tourism experiences are primarily focused on traditional sightseeing, with few opportunities for in-depth participation. Special ICH cultural festivals and folk activities, such as singing, dancing, performing, drama, and juggling, should be organized for different tourist audience groups, allowing visitors to gain an in-depth understanding of local folk customs and the spirit of Chinese tradition through rural ICH tours. Creating an ICH cultural IP and optimizing the destination image are crucial. The key goal of creating a successful destination image and commercializing it is to align tourists’ perception of the destination with the image projected by marketers [42]. Therefore, in day-to-day management and operations, ICH culture should be used as the leading tourism resource to guide tourists in forming a perception of the destination image, to enhance the competitiveness of rural ICH tourism, and to foster tourist loyalty. To optimize tourists’ perception of the rural ICH tourism destination, it is suggested that three aspects should be focused on: rural environment management, image creation, and promotion. Tourists’ perception of rural ICH tourism with regard to aspects such as the experiential environment, leisure environment, cultural environment, auxiliary facilities, and service quality, can be improved by creating an ICH cultural IP, organizing rural ICH tourism cultural festivals, regularly holding ICH cultural forums and exchanges, and developing cultural and creative products;In the effective transition from poverty alleviation to rural revitalization, the development of rural intangible heritage tourism must confront existing issues and shortcomings in its initial stages, break through development bottlenecks, and achieve high-quality growth in rural intangible heritage tourism. In facing issues such as insufficient motivation, low quality, and poor benefits in rural ICH tourism, maintaining the authenticity of rural ICH projects, enhancing tourist engagement and interaction, and establishing a positive tourism image are effective measures for integrating rural ICH tourism with rural revitalization. By adopting characteristic, differentiated, personalized, and diversified development paths and models for rural ICH tourism, enriching and perfecting product structures and services, strengthening the image of tourist destinations, revitalizing the use of rural ICH projects, and employing modern technologies, the internet, branding, and capital, rural ICH projects can continue to thrive, driving the diversified transformation and sustainable development of the rural tourism economy.

### 5.3. Limitations and Future Research Directions

The limitations of this study are as follows: firstly, there is a certain limitation in the sample selection. The data originate from Meizhou Island, a typical destination for ICH tourism, but different types of ICH and destination types vary in their attractiveness to tourists. Future research could select other types of rural ICH tourism destinations to verify the generalizability of the conclusions. Furthermore, the sampling method of this study may impact the representativeness of the results. Although this method increases the convenience and efficiency of sample selection, it may also limit the universality of the findings. Future research is advised to employ more representative sampling methods to verify and expand upon the current findings. On the other hand, there is a limitation in the measurement of the variables. The scales for perceived authenticity and cognitive destination image needed to be adjusted according to the actual conditions of the case study area. Thus, a degree of subjectivity was introduced. Future research could adopt a mixed-method approach and develop suitable scales based on specific tourist destinations to enhance the adaptability and objectivity of the scales.

## Figures and Tables

**Figure 1 behavsci-14-00302-f001:**
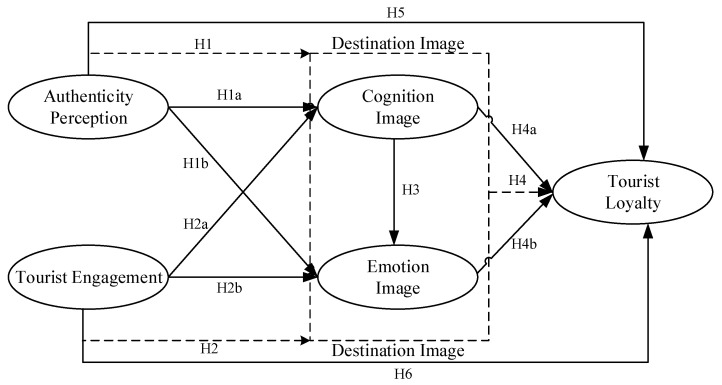
Research hypothesis model.

**Figure 2 behavsci-14-00302-f002:**
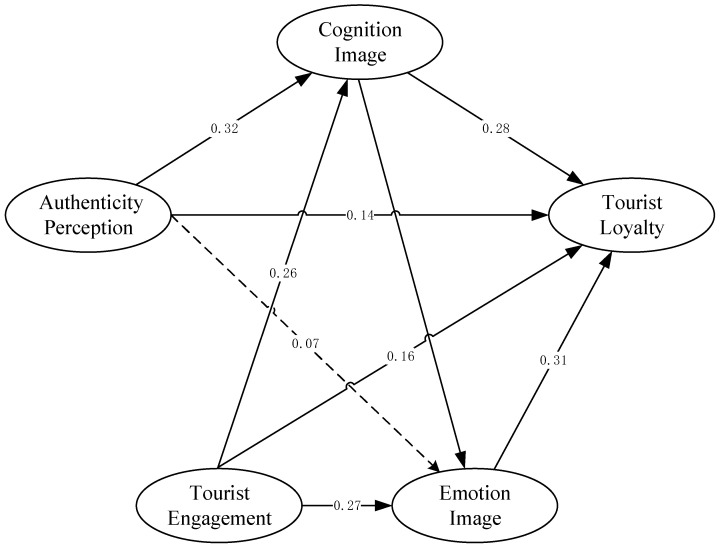
The revised SEM model. Dashed lines indicate non-significant paths that have been removed.

**Table 1 behavsci-14-00302-t001:** Demographic characteristics and basic statistics of the sample.

Item	Category	Frequency	Percentage (%)
Frequency of Visits	First Time	252	54
	Second Time	79	16.9
	Third Time	24	5.1
	More than three times	112	24
Education Level	Junior High School and below	90	19.3
	Senior High School	123	26.3
	Associate Degree (Diploma)	82	17.6
	Bachelor’s Degree	130	27.8
	Postgraduate Degrees	42	9
Monthly Income (RMB)	≤3000	105	22.5
	3001–6000	120	25.7
	6001–9000	91	19.5
	9001–12,000	60	12.8
	≥12,001	91	19.5
Gender	Male	231	49.5
	Female	236	50.5
Age Group	≤15 years	12	2.6
	16–25 years	115	24.6
	26–35 years	171	36.6
	36–45 years	80	17.1
	46–55 years	52	11.1
	56–65 years	32	6.9
	≥66 years	5	1.1
Level of Understanding of Mazu Culture	Completely unfamiliar	27	5.8
	Have heard of it but do not know much	115	24.6
	Somewhat familiar	180	38.5
	Fairly knowledgeable	73	15.6
	Extremely knowledgeable	72	15.4

**Table 2 behavsci-14-00302-t002:** Reliability and validity test.

Factor (Cronbach’s α)		λ	CR	AVE
Criterion		>0.5	>0.7	>0.5
Perceived Authenticity (0.892)	Experienced the authentic culture of Mazu belief in its original form.	0.784	0.894	0.546
	A strong atmosphere characteristic of Mazu belief culture.	0.721		
	Sensed the devotional spirit of followers of Mazu culture.	0.714		
	Observed the devout behaviors or activities of Mazu culture adherents.	0.665		
	Mazu temples and statues are well preserved.	0.731		
	An excellent representation of the real Mazu belief culture (objective fact).	0.753		
	Enabled me to experience the real Mazu belief culture (subjective feeling).	0.797		
Tourist Engagement (0.877)	Utilized guide tools during the tour.	0.669	0.878	0.51
	Sought assistance from local residents when encountering problems during the tour.	0.66		
	Interacted with local residents during the tour.	0.614		
	Visited sites of Mazu belief culture such as ancestral Mazu temples and Mazu cultural parks.	0.765		
	Participated in local Mazu cultural folk performances or activities.	0.744		
	Tasted local special foods of Mazu culture.	0.759		
	Employed smart devices to learn about Mazu culture.	0.769		
Cognitive Image—Experiential Environment (0.812)	The local environment is secure.	0.791	0.815	0.595
	The local environment is clean, tidy, and comfortable.	0.702		
	Local residents are friendly and helpful.	0.817		
Cognitive Image—Leisure Shopping (0.802)	Special dining/cuisine is diverse and delicious.	0.795	0.808	0.584
	Shops/memorabilia are abundant and unique.	0.756		
	A wide selection of hotels and guesthouses is available nearby.	0.74		
Cognitive Image—Cultural Environment (0.853)	The area has unique cultural heritage and history.	0.755	0.858	0.548
	Architectural/temple cultural styles are unique and authentic.	0.717		
	Rich in distinctive performances, exhibitions, or activities.	0.677		
	A unique and rich folk culture worth visiting.	0.768		
	The intangible cultural heritage of Mazu belief is well protected and developed.	0.78		
Cognitive Image—Auxiliary Facilities (0.774)	Local facilities are good (signage, maps, information screens, etc.).	0.827	0.787	0.554
	Comprehensive and convenient online information services, such as websites and public accounts.	0.71		
	Specialized scenic area service management/tourist reception centers.	0.688		
Cognitive Image—Value and Services (0.832)	Reasonable prices for local food and accommodation.	0.771	0.833	0.625
	Reasonable prices for attraction tickets and activities.	0.762		
	Staff with extensive professional knowledge and good attitudes.	0.836		
Affective Image (0.815)	Either displeasing or comfortable.	0.8	0.818	0.6
	Either soporific or exhilarating.	0.738		
	Either melancholic and dull or exciting.	0.784		
Tourist Loyalty (0.867)	Given the chance, I would like to visit Meizhou Island again.	0.763	0.871	0.576
	I plan to revisit Meizhou Island in the future.	0.704		
	I would recommend Meizhou Island to my relatives and friends.	0.742		
	I give positive word-of-mouth accounts and reviews about my trip to Meizhou Island.	0.768		
	Given the opportunity, I am willing to visit other rural intangible cultural heritage tourism destinations.	0.812		

**Table 3 behavsci-14-00302-t003:** Initial path coefficient.

Path			Standardized Coefficient	Unstandardized Coefficient	S.E.	C.R.	*p*	Test
								Result
Cognitive Image	<---	Loyalty–Authenticity Perception	0.316	0.258	0.047	5.538	***	Supported
Affective Image	<---	Loyalty–Authenticity Perception	0.068	0.076	0.064	1.179	0.238	Not Supported
Cognitive Image	<---	Tourist Engagement	0.259	0.21	0.046	4.538	***	Supported
Affective Image	<---	Tourist Engagement	0.254	0.28	0.065	4.298	***	Supported
Affective Image	<---	Cognitive Image	0.287	0.39	0.085	4.578	***	Supported
Loyalty	<---	Loyalty–Authenticity Perception	0.136	0.128	0.047	2.714	0.007	Supported
Loyalty	<---	Tourist Engagement	0.156	0.145	0.048	3.007	0.003	Supported
Loyalty	<---	Cognitive Image	0.285	0.326	0.066	4.947	***	Supported
Loyalty	<---	Affective Image	0.313	0.264	0.047	5.647	***	Supported

Note. *** indicates *p* < 0.001.

**Table 4 behavsci-14-00302-t004:** Path coefficient of SEM model.

Path			Standardized Coefficient	Unstandardized Coefficient	S.E.	C.R.	*p*	Test
								Result
Cognitive Image	<---	Authenticity Perception	0.319	0.261	0.047	5.599	***	Supported
Cognitive Image	<---	Tourist Engagement	0.257	0.209	0.046	4.508	***	Supported
Affective Image	<---	Tourist Engagement	0.271	0.3	0.064	4.714	***	Supported
Affective Image	<---	Cognitive Image	0.312	0.426	0.081	5.244	***	Supported
Loyalty	<---	Authenticity Perception	0.14	0.131	0.047	2.777	0.005	Supported
Loyalty	<---	Tourist Engagement	0.156	0.145	0.049	2.982	0.003	Supported
Loyalty	<---	Cognitive Image	0.284	0.325	0.067	4.877	***	Supported
Loyalty	<---	Affective Image	0.313	0.263	0.047	5.644	***	Supported

Note. *** indicates *p* < 0.001.

**Table 5 behavsci-14-00302-t005:** Mediation effect test.

Path	Standardization	Bias-Corrected		Percentile	
	Effect Size	95%CI		95%CI	
		Lower Bound	Upper Bound	Lower Bound	Upper Bound
Total Effect					
Authenticity Perception → Loyalty	0.261	0.152	0.373	0.154	0.375
Tourist Engagement → Loyalty	0.339	0.234	0.454	0.231	0.448
Indirect Effect					
Authenticity Perception → Cognitive Image → Loyalty	0.091	0.045	0.163	0.039	0.152
Authenticity Perception → Cognitive Image →Affective Image → Loyalty	0.031	0.013	0.069	0.01	0.062
Tourist Engagement → Cognitive Image → Loyalty	0.073	0.035	0.145	0.029	0.13
Tourist Engagement → Affective Image → Loyalty	0.085	0.036	0.156	0.03	0.147
Tourist Engagement → Cognitive Image → Affective Image → Loyalty	0.025	0.01	0.055	0.008	0.049
Direct Effect					
Authenticity Perception → Loyalty	0.14	0.037	0.239	0.044	0.245
Tourist Engagement → Loyalty	0.156	0.065	0.252	0.066	0.253

## Data Availability

The raw data supporting the conclusions of this manuscript will be made available by the authors to any qualified researcher.

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
