# Peer review of "The Post-Effects of the Authenticity of Rural Intangible Cultural Heritage and Tourists’ Engagement"

_behavsci, 2024, doi:10.3390/bs14040302_

Round 1

Reviewer 1 Report

Comments and Suggestions for Authors

Although I found this paper quite well-written, the only concern for me is that this paper is more of a tourism research than a "behavioral" research.

Regardless of those mentioned above, my comments are as follows:

1. it is important to present the process of convenient sample selection and sampling procedure for onsite and online samplings.  

2. Apparently, the sample bias is quite salient in this study.  The sample comprised of majority respondents who are first-timers, aged 35 or younger, and with an income that is 6,000 or under.  Such bias may affect the generalizability of the study results and thus should be addressed.  

and I believe it is very critical to address this issue in this study.  Furthermore, 

Comments on the Quality of English Language

minor revision of English is needed.

Author Response

Dear reviewers:

Thank you for your suggestions from which we have benefited immensely. We have revised this manuscript according to your suggestions and now believe that the article has improved in terms of logic and fluency. We used revision tracking.

Suggestion 1:

Although I found this paper quite well-written, the only concern for me is that this paper is more of a tourism research than a "behavioral" research.

Response: Thanks for your suggestions. This study was submitted to the "The Emotional Antecedents and Consequences of Buying and Consuming: A Multidisciplinary Perspective on Consumers’ Emotions" Special Issue. Our study investigates the behavioral decisions, psychological states, or behavioral patterns of tourists during travel activities, including variables related to emotions.  Therefore, this research aligns with the thematic scope of this special issue call for papers.

Suggestion 2:

it is important to present the process of convenient sample selection and sampling procedure for onsite and online samplings.  

Response: Thanks for your suggestions. Thanks for your suggestions. Following your requests, we have revised the section to provide a more detailed introduction to the questionnaire distribution form and specific steps. In section 5.3, Limitations and Future Research Directions, we have discussed the limitations of our sampling method.

Suggestion 3:

  1. Apparently, the sample bias is quite salient in this study.  The sample comprised of majority respondents who are first-timers, aged 35 or younger, and with an income that is 6,000 or under.  Such bias may affect the generalizability of the study results and thus should be addressed.  

and I believe it is very critical to address this issue in this study.  Furthermore,

Response: Thank you for your detailed comments on the issue of sample bias in our research. We acknowledge the presence of biases in terms of age, proportion of first-time visitors, and income levels in our sample. These biases largely reflect the actual situation of our research setting, the tourist structure of Meizhou Island in Fujian Province. 1) According to official data released by the Fujian Provincial Government, the average income level is 45,400 RMB per year, which coincides with the majority of our sample's income level being below 6,000 RMB. 2) Most tourists do not frequently visit the same scenic area, especially since Meizhou Island is far from urban areas. Non-local tourists may need to change three types of transportation to reach the island, taking a train or plane to Putian City, then a bus or taxi to the pier, and finally a ferry to the island. 3) Young people, as the main force of tourism, are also more willing to participate in surveys, which is reflected in other studies. We understand that sample bias may affect the generalizability of our research findings. However, overall, in terms of education level, age, and understanding of Mazu culture, the sample generally follows a normal distribution. Nevertheless, we agree that sample bias is an issue that requires further attention. Therefore, in section 5.3, Limitations and Future Research Directions, we propose the use of broader sampling methods in the future to enhance the representativeness of the sample. Thank you again for your valuable suggestions.

Suggestion 4:

minor revision of English is needed.

Response: Thanks for your suggestions. We have sought MDPI's English Editing service for further refinement, order number: 76873.

Reviewer 2 Report

Comments and Suggestions for Authors

Tittle: The Post-effects of the Authenticity of Rural Intangible Cultural Heritage and Tourists' Engagement

In the abstract: “The results show that: 13 1) Tourists' perception of authenticity has a significant positive impact on the cognitive image of the 14 destination but has no impact on the affective image of the destination, and visitor engagement has 15 a significant positive impact on destination cognition and affective image; 2) …..” why authors gave such points, it should be rewrite as “normal” text, and goals of the article.

Also every studies should started from explanation of the main definitions, and here the authors didn’t give them.

In lines: 141-142: Destination image is one of the key concepts in the field of tourism destination marketing and an important benchmark for market positioning and the competitive analysis  of tourism destinations – on that level is not need to point the benchmark.

Used sources: Oliver, R.L. Whence consumer loyalty? J Marketing 1999, 63, 33-44. 640 22. Wang, L.; Li, X. The five influencing factors of tourist loyalty: A meta-analysis. Plos One 2023, 18, e283963.  – are too old – especially after COVID-19 it is need to use more present information and sources.  

The research part is good organized so they don’t have to be rewrite.

Author Response

Dear reviewers:

Thank you for your suggestions from which we have benefited immensely. We have revised this manuscript according to your suggestions and now believe that the article has improved in terms of logic and fluency. We used revision tracking.

Suggestion 1:

In the abstract: “The results show that: 13 1) Tourists' perception of authenticity has a significant positive impact on the cognitive image of the 14 destination but has no impact on the affective image of the destination, and visitor engagement has 15 a significant positive impact on destination cognition and affective image; 2) …..” why authors gave such points, it should be rewrite as “normal” text, and goals of the article.

Response: Thanks for your suggestions. As per your request, we have rewritten the abstract, emulating the style of other behavioral articles.

Suggestion 2:

Also every studies should started from explanation of the main definitions, and here the authors didn’t give them.

Response: Thanks for your suggestions. Per your instruction, we have added definitions of various variables in the Introduction section.

Suggestion 3:

In lines: 141-142: Destination image is one of the key concepts in the field of tourism destination marketing and an important benchmark for market positioning and the competitive analysis  of tourism destinations – on that level is not need to point the benchmark.

Response: Thanks for your suggestions. We have reduced and rephrased this part of the content.

Suggestion 4:

Used sources: Oliver, R.L. Whence consumer loyalty? J Marketing 1999, 63, 33-44. 640 22. Wang, L.; Li, X. The five influencing factors of tourist loyalty: A meta-analysis. Plos One 2023, 18, e283963.  – are too old – especially after COVID-19 it is need to use more present information and sources.  

Response: Thanks for your suggestions. Following your advice, we have added relevant references, replaced some outdated ones, and included some materials published after 2023. However, some references related to the origin of scales or variables could not be replaced.